# Age-related changes in the neuromuscular control of forward and backward locomotion

**Arthur H. Dewolf**[1]*, **Francesca Sylos-Labini**[2], **Germana Cappellini**[2,3], **Yury Ivanenko**[2], **Francesco Lacquaniti**[1,2]

**1** Department of Systems Medicine and Center of Space Biomedicine, University of Rome Tor Vergata, Rome, Italy, **2** Laboratory of Neuromotor Physiology, IRCCS Santa Lucia Foundation, Rome, Italy, **3** Department of Pediatric Neurorehabilitation, IRCCS Santa Lucia Foundation, Rome, Italy

\* arthur.dewolf@uclouvain.be

**Data Availability Statement:** All relevant data are within the manuscript and its Supporting Information files.

## Abstract

Previous studies found significant modification in spatiotemporal parameters of backward walking in healthy older adults, but the age-related changes in the neuromuscular control have been considered to a lesser extent. The present study compared the intersegmental coordination, muscle activity and corresponding modifications of spinal montoneuronal output during both forward and backward walking in young and older adults. Ten older and ten young adults walked forward and backward on a treadmill at different speeds. Gait kinematics and EMG activity of 14 unilateral lower-limb muscles were recorded. As compared to young adults, the older ones used shorter steps, a more in-phase shank and foot motion, and the activity profiles of muscles innervated from the sacral segments were significantly wider in each walking condition. These findings highlight age-related changes in the neuromuscular control of both forward and backward walking. A striking feature of backward walking was the differential organization of the spinal output as compared to forward gait. In addition, the resulting spatiotemporal map patterns also characterized age-related changes of gait. Finally, modifications of the intersegmental coordination with aging were greater during backward walking. On the whole, the assessment of backward walk in addition to routine forward walk may help identifying or unmasking neuromuscular adjustments of gait to aging.

## Introduction

Age-related changes in the gait features of forward locomotion have been studied extensively [1–4], and have been related to risk of falling [5] and/or to health status in older adults [6]. The modifications of walking between young and older adults occur in parallel with, amongst other factors, changes in the musculoskeletal system [7,8] as well as the central and peripheral nervous systems [9,10]. A number of studies have provided new insights about the plasticity of the neuromuscular control of gait to adapt to those age-related physiological changes (e.g., [11–13]).

In particular, a distal-to-proximal redistribution of joint efforts has been established as a gait feature of older adults [14]. Part of this so-called biomechanical plasticity has been related

**Funding:** This work was supported by the Italian Ministry of Health (Ricerca corrente, IRCCS Fondazione Santa Lucia), Italian Space Agency (grants I/006/06/0 and ASI-MARS-PRE DC-VUM - 2017-006), the H2020-779963 EUROBENCH FSTP-1 grant (sub-project PEPATO), and Italian University Ministry (PRIN grant 2017CBF8NJ_005). The funders had no role in study design, data collection and analysis, decision to publish, or preparation of the manuscript.

**Competing interests:** The authors have declared that no competing interests exist.

**Abbreviations:** EMG, electromyography; FWHM, full width at half maximum; MN, motoneurons; $\eta p2$, eta square; $u_i$, eigenvectors; PV, percentage of variance explained by eigenvectors; ROM, range of motion.

to a reduction of muscle strength [12,15–17]. However, other studies have revealed that older adults may retain the muscular potential to develop higher ankle power, and in turn reduce the distal-to-proximal redistribution [18] under specific situations (*e.g.*, during walking uphill [19] or using biofeedback [20]). The decline of propulsive power generation during push-off is thus not only due to a reduced muscular capacity but might also emerge from a different neuromuscular control strategy [19]. Therefore, recent efforts have been made to understand the age-related plasticity of the neuromuscular system during forward walking [1,13,21–23]. For example, using the planar covariation law [24,25], significant adjustments of the intersegmental coordination related to aging, and especially of the shank-foot coordination, have been documented. In addition, aging also involves modifications of muscle activations [26–29], suggesting a loss of fine neural control [29,30]. These changes are reflected at the level of the alpha-motoneuronal (MN) activity [28], however the modifications of spinal locomotor output have never been quantified.

Although continuous backward walking occurs rarely, it is still critical for independence in daily life, *e.g.*, when stepping back in front of a forthcoming vehicle, when opening a door, when backing up to sit down [31]. Backward walking has been extensively studied in the context of theories on the organization of central pattern generators (CPGs). As first hypothesized by Grillner [32], it has been suggested that backward walking is basically forward in reverse [33–36]. While the kinematics seems to support this idea [36], suggesting sharing circuitry [35,37], such reversal is not present for muscle activity, especially for ankle muscles [33,36]. Ivanenko et al. [38] have also noted important differences in the spinal cord MN activity between backward and forward walking, suggesting a partial reconfiguration of lower level networks [39]. In addition, backward walking requires the involvement of specialized control circuits [40] mainly at supraspinal levels [41], suggesting that it is more challenging to the nervous system than standard forward walking.

For this reason, there has been a growing interest in the use of backward walking for rehabilitation purposes. Recent studies suggest that backward walking can be used for rehabilitation or for diagnostics in patients with neurological injuries [42–47]. Since older adults rely more on visual feedback during both standing and walking than young adults [48–51], it has been hypothesized that backward gait may be used to unmask mobility impairments and assess risk of falling. Compared to young adults, backward walking in older adults is characterized by higher stride frequency, slower speed, and increased gait variability [2,31,52]. However, perhaps due to subtle neuromuscular adjustments associated with normal aging [22,28,53], it is still unclear how the neuromuscular control adapts to backward walking with aging.

To the best of our knowledge, the present study is the first to provide quantitative comparisons of the pattern generator output during forward and backward walking between young and older adults. We intended to better pinpoint underlying mechanisms of age-related neuromuscular adaptations in both backward and forward walking. In particular, because backward walking is more challenging than forward walking and because patterns of neuromuscular control are direction specific in humans [54], we wondered whether backward walking can reveal age-related modifications of gait that are not otherwise apparent during forward walking.

Altered spatiotemporal stride parameters [2,31], altered coordination patterns among the elevation angles of the lower limb segments [1,13], and wider bursts of muscle activity [28,29] have been previously documented for the forward locomotion of older adults. Here, we expected that some of these alterations might apply also to backward walking. In particular, we expected age-related adjustments of the intersegmental coordination, namely a more in-phase shank and foot motion, as well as a widening of muscle activities. Importantly, we also expected that some of these age-related modifications might be reflected in the pattern of

rostrocaudal activation of the motoneuron pools. Finally, we hypothesized that these age-related differences of neuromuscular control would be more pronounced during backward walking compared with forward walking.

## Methods

### Subject and experimental procedure

Ten young (4 ♀; age: 28.7±5.1 yrs, mass: 74.5±10.7 kg, height: 1.75±0.07 m, means±SD) and 10 older adults (1 ♀; age: 73.5±4.5 yrs, mass: 81.5±5.9 kg, height: 1.76±0.05 m, mean±SD) participated to the study. Mass and height were not significantly different between young and older adults (mass: $t = 0.5$; $p = 0.605$; height: $t = 1.8$; $p = 0.086$). The number of subjects was determined by a priori power analysis using the G*Power program. Based on the age-related difference of stride length during forward walking on a treadmill [1], a total sample of 18 participants (9 per group) would be sufficient to detect a large effect size ($\eta_p^2 = 0.20$) with 90% power, using one-way ANOVA with $p = 0.05$. No subject had a recent history of falling. All participants were able to walk without assistance and did not complain about musculoskeletal disorders. All participants gave written informed consent. Experiments were performed according to the Declaration of Helsinki and were approved by the ethics committee of IRCCS Santa Lucia Foundation (CE/PROG749).

Participants were asked to walk forward and backward while they wore their own walking shoes on a treadmill at two different fixed speeds [2 (0.56) and 3 (0.83) km h$^{-1}$(m s$^{-1}$) backward, 2 (0.56) and 4 (1.11) km h$^{-1}$ (m s$^{-1}$) forward]. Two young adults did not perform the walking tasks at 2 km h$^{-1}$ and one older adult was unable to walk backward at 3 km h$^{-1}$. During backward walking, subjects were allowed to hold a hand rail with their left hand for balance. For each trial, at least 8 consecutive strides were analysed (Table 1). Bilateral, full-body three-dimensional (3D) kinematics was recorded at 200 Hz by means of a Vicon-612 system (Oxford, UK) with nine cameras placed around the treadmill. Twelve reflective markers were attached to the skin of the subjects overlying the following bilateral landmarks: gleno-humeral joint, lateral epicondyle of the elbow, ulnar process of the wrist, greater trochanter, lateral femur epicondyle and lateral malleolus. In addition, four markers were placed on each shoe in approximate correspondence with the heel, and fifth metatarso-phalangeal joint. The EMG data were recorded at 2000 Hz by means of a Delsys Trigno Wireless System (Boston, MA). The following 14 muscles were recorded on the right side of the body: *erector spinae* (*ES*) at L2 level, *gluteus maximus* (*Gmax*), *gluteus medius* (*Gmed*), *tensor fasciae latae* (*TFL*), *vastus*

**Table 1. Number of muscles (and number of strides) analysed per subject in each walking condition (e.g. FW 2 is forward walk at 2 km h$^{-1}$).**

| | Young | | | | | Older | | | |
|---|---|---|---|---|---|---|---|---|---|
| *Subject* | *FW 2* | *FW 4* | *BW 2* | *BW 3* | *Subject* | *FW 2* | *FW 4* | *BW 2* | *BW 3* |
| Y1 | 14 (12) | 14 (16) | 12 (17) | 13 (16) | E1 | 12 (11) | 11 (10) | 11 (10) | 11 (8) |
| Y2 | 13 (12) | 14 (12) | 14 (13) | 14 (17) | E2 | 14 (11) | 14 (15) | 14 (10) | 14 (8) |
| Y3 | 14 (13) | 14 (12) | 13 (12) | 13 (13) | E3 | 13 (12) | 13 (11) | 11 (12) | 13 (9) |
| Y4 | 8 (10) | 10 (11) | 8 (11) | 8 (11) | E4 | 13 (12) | 13 (11) | 13 (11) | 13 (8) |
| Y5 | 13 (11) | 13 (12) | 13 (11) | 13 (11) | E5 | 14 (13) | 13 (17) | 14 (15) | 14 (15) |
| Y6 | 12 (10) | 12 (11) | 11 (11) | 13 (10) | E6 | 7 (11) | 8 (10) | 12 (11) | 10 (10) |
| Y7 | 14 (11) | 11 (12) | 12 (10) | 13 (12) | E7 | 9 (14) | 9 (12) | 12 (11) | 11 (10) |
| Y8 | 13 (13) | 13 (16) | 13 (13) | 13 (11) | E8 | 14 (13) | 13 (11) | 13 (11) | 10 (10) |
| Y9 | – | 13 (10) | – | 14 (8) | E9 | 13 (12) | 12 (13) | 13 (15) | 13 (19) |
| Y10 | – | 14 (9) | – | 14 (8) | E10 | 13 (10) | 10 (10) | 12 (11) | – |

*medialis* (*VM*), *vastus lateralis* (*VL*), *rectus femoris* (*RF*), long head of the *biceps femoris*, (*BF*), *semitendinosus* (*ST*), *tibialis anterior* (*TA*), *medial gastrocnemius* (*MG*), *lateral gastrocnemius* (*LG*), *soleus* (*SOL*) and *peroneus longus* (*PERL*). EMG electrodes were placed based on suggestions from SENIAM (*seniam.org*), the European project on surface EMG. To ensure correct placement of EMG electrodes, muscle bellies were located by means of palpation and the electrodes were oriented along the main direction of the fibers [55]. In certain conditions, some electrodes became partially detached and the data series produced by these electrodes were removed from the analysis (replaced by a not-a-number vector) on a subject-specific basis (S1 Table). Table 1 presents the number of muscles and strides analysed for each subject in each walking condition. Kinematic and EMG recordings were synchronized on-line. All analyses were performed using custom Matlab sofware (MathWorks Inc., MA, USA).

## Kinematic data analysis

The stride was defined as the period between two ground contacts of the right foot. Foot-contact was estimated according to the local minima of the vertical displacement of the heel marker [56], while the timing of the lift-off was estimated from the maximum excursion of the lower limb elevation angle, defined as the angle between the vertical axis and the whole limb segment (from the greater trochanter to the lateral malleolus), projected on the sagittal plane [24].

From the marker locations, the orientation of the thigh, shank, foot and trunk relative to the vertical axis (elevation angle) were computed as described in Borghese et al. [24]. For each participant, the duration of different strides of each trial was normalized by interpolating individual gait cycles over 200 points. To analyse the relative phase of the time-course of the elevation angles during a stride, the phase lags between two adjacent limb-segments were computed by means of cross-correlation function.

As reported in prior studies, a principal component analysis was applied to determine the covariance matrix of the segment elevation angles [57], after subtraction of the mean value. Notice that, for this analysis, the amplitude of these angles was not normalized. Eigenvalues and eigenvectors $u_i$ were computed by factoring the covariance matrix from the set of original signals by means of a singular value decomposition algorithm. The first two eigenvectors ($u_1$ and $u_2$) lied on the best-fitting plane of angular covariation, and the data projected onto the corresponding axes corresponded to the first ($PC_1$) and second ($PC_2$) principal components. The planarity was evaluated for each condition by calculating the percentage of variance that was explained by $u_1$ ($PV_1$), $u_2$ ($PV_2$) and $u_3$ ($PV_3$). If the data lay perfectly on a plane, $PV_1$ + $PV_2$ would be 100% (and $PV_3$ would be 0%). By definition, the third eigenvector $u_3$ is orthogonal to the plane defined by $u_1$ and $u_2$. The parameter $u_{3t}$ corresponds to the direction cosine with the positive semi-axis of the thigh and provides one measure of the orientation of the plane.

## EMG data analysis

The collected raw EMG signals were high-pass filtered (30 Hz), then rectified and low-pass filtered with a zero-lag third-order Butterworth filter (10 Hz). As for the kinematic data, the time scale was normalized by interpolating individual gait cycles over 200 points. For each condition and for each EMG (rectified, filtered) waveform, the full width at half maximum (FWHM) was calculated as the period during which the EMG activity exceeded the half of its maximum [29,58,59].

The EMG activities were normalized to unit variance across all trials [60] and then mapped onto the estimated rostro-caudal location of the MN pools in the human spinal cord from the

L2 to S2 segments based on Kendall's myotomal charts [55], as in Ivanenko et al [61,62]. To account for size differences in MN pools at each spinal level, this fractional activity value was then multiplied by the estimated segment-specific number of MNs ($MN_j$), based on Tomlinson and Irving [63]. Note that, consistent with previous work [64], the spinal maps were relatively insensitive to the subset of muscles analysed (Table 1). Indeed, spinal maps reconstructed from a subset of seven muscles (minimum number of muscles recorded) were strongly correlated with the maps computed from the full set of muscles, with average correlation coefficients between 0.9–0.99 for each task and at each individual spinal segment (S2 Table).

To compute the relative activation of the lumbar and sacral segments in each condition, we averaged the motor output patterns over the gait cycle in the upper part of the lumbar segments (sum of the activity from L2 to L4) and the sacral segments (sum of activity from S1 to S2). To reduce overlaps due to maps smoothing, the spinal segment L5 was not taken into account [65,66]. The FWHM, the maximal activation, and its timing were calculated for both lumbar and sacral segments.

## Statistics

The statistical analysis was designed to assess the effect of progression speed, direction (backward *vs*. forward), age group (young *vs*. older), and the interaction between these factors. A general linear mixed model was applied, with the direction and speed defined as repeated measures. The normality of the residuals was checked by means of the Kolmogorov-Smirnov test. Normality was not assumed for 5 variables (range of motion of the trunk elevation angle, $PV_3$, range of motion of the thigh elevation angle and FWHM of *Gmax* and *TFL*). In these cases, an inverse (for the parameter trunk and thigh ROM) or log (for the other parameters) transform was applied, and the normality of the residuals was then assumed. In each Figure, the asterisks indicate significant student t-tests with Benjamini-Hochberg *p-level* adjustment [67] comparing the age groups. The effect size, measure by the eta square ($\eta_p^2$), is reported for age group comparisons.

## Results

### Gait and kinematic parameters

The stride period (and stride length) decreased with speed ($F_{2,67}$ = 32.3; $p$ < 0.001; Fig 1C) and was significantly affected by both the age and the direction of progression: the stride period decreased when walking backward as compared to walking forward ($F_{1,67}$ = 18.7; $p$ < 0.001), and in older as compared to young adults ($F_{1,67}$ = 71.1; $p$ < 0.001; $\eta_p^2$ = 0.49). The relative duration of the stance phase was shorter with increasing speeds ($F_{2,67}$ = 17.6; $p$ < 0.001) and in older adults ($F_{1,67}$ = 32.3; $p$ < 0.001; $\eta_p^2$ = 0.34), but was not significantly affected by the direction of progression ($F_{1,67}$ = 2.1; $p$ = 0.155).

When speed increased in both backward and forward walking, the range of motion (ROM) of the thigh, shank and foot elevation angles increased (thigh: $F_{2,67}$ = 8.2; $p$ = 0.001; shank: $F_{2,67}$ = 63.9; $p$ < 0.001; foot: $F_{2,67}$ = 37.7; $p$ < 0.001) in both groups (Fig 1A and 1B), whereas the ROM of the trunk elevation angle and its mean inclination were not significantly affected (ROM: $F_{2,67}$ = 0.8; $p$ = 0.476; mean: $F_{2,67}$ = 1.6; $p$ = 0.203). During backward walking, the thigh, shank and foot ROM significantly decreased (thigh: $F_{2,67}$ = 4.8; $p$ = 0.032; shank: $F_{1,67}$ = 26.6; $p$ < 0.001; foot: $F_{1,67}$ = 57.9; $p$ < 0.001), the trunk ROM increased ($F_{1,67}$ = 8.9; $p$ = 0.006), whereas the trunk mean inclination was not significantly different relative to forward walking (trunk: $F_{1,67}$ = 1.2; $p$ = 0.289).

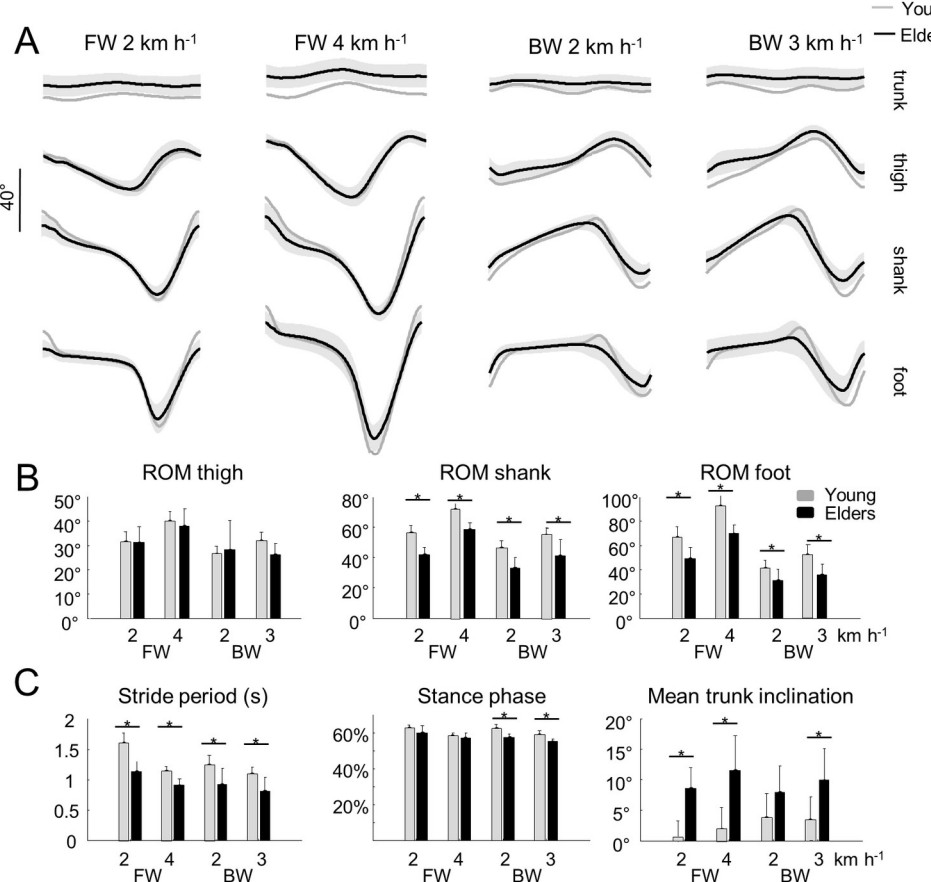

**Fig 1. Elevation angles of lower-limb segments and general gait parameters during forward (FW) and backward (BW) walking in young and older adults.** *A*—Elevation angles of the trunk, thigh, shank and foot over a stride. All the curves of each subject walking at a given walking condition were first averaged (mean-curve). The curves presented here are the average of the mean-curves of all the young (grey lines) and older (black lines) adults. The grey zone represents±1 SD for the older adults. *B*—Average range of motion of the thigh, shank and foot over one stride. *C*—Average, stride period, relative stance phase and mean trunk inclination over one stride. In panel B and C, the bars represent the grand mean of all the young (grey) and the older (black) adults. Thin lines represent one standard deviation. The * indicates a significant effect of age.

In both age groups, the time-varying waveform of the elevation angles remained fairly similar across walking conditions (Fig 1A). However, the ROM of the shank and the foot segments were significantly smaller (shank: $F_{1,67}$ = 87.0; $p <$ 0.001; $\eta_p^2$ = 0.58; foot: $F_{1,67}$ = 75.1; $p <$ 0.001; $\eta_p^2$ = 0.53), and the mean trunk inclination was significantly greater ($F_{1,67}$ = 51.7; $p <$ 0.001; $\eta_p^2$ = 0.45) in older than in young adults. The trunk and thigh ROM were not significantly different between young and older adults (trunk: $F_{1,67}$ = 1.2; $p$ = 0.288; thigh: $F_{1,67}$ = 0.617; $p <$ 0.543).

## Intersegmental coordination

The coordination between thigh, shank and foot elevation angles was evaluated using principal component analysis (Fig 2). Fig 2A illustrates the averaged gait loops plotted in 3D during backward and forward walking in both age groups. Notice the appreciably smaller loops in backward walking and in older adults (along with smaller ROM).

In each condition, $PV_1 + PV_2 >$ 97% (Fig 2B). However, $PV_1$ was significantly smaller (and $PV_2$ greater) in older than in young adults ($PV_1$: $F_{1,67}$ = 23.7; $p <$ 0.001; $\eta_p^2$ = 0.26; PV2: $F_{1,67}$ =

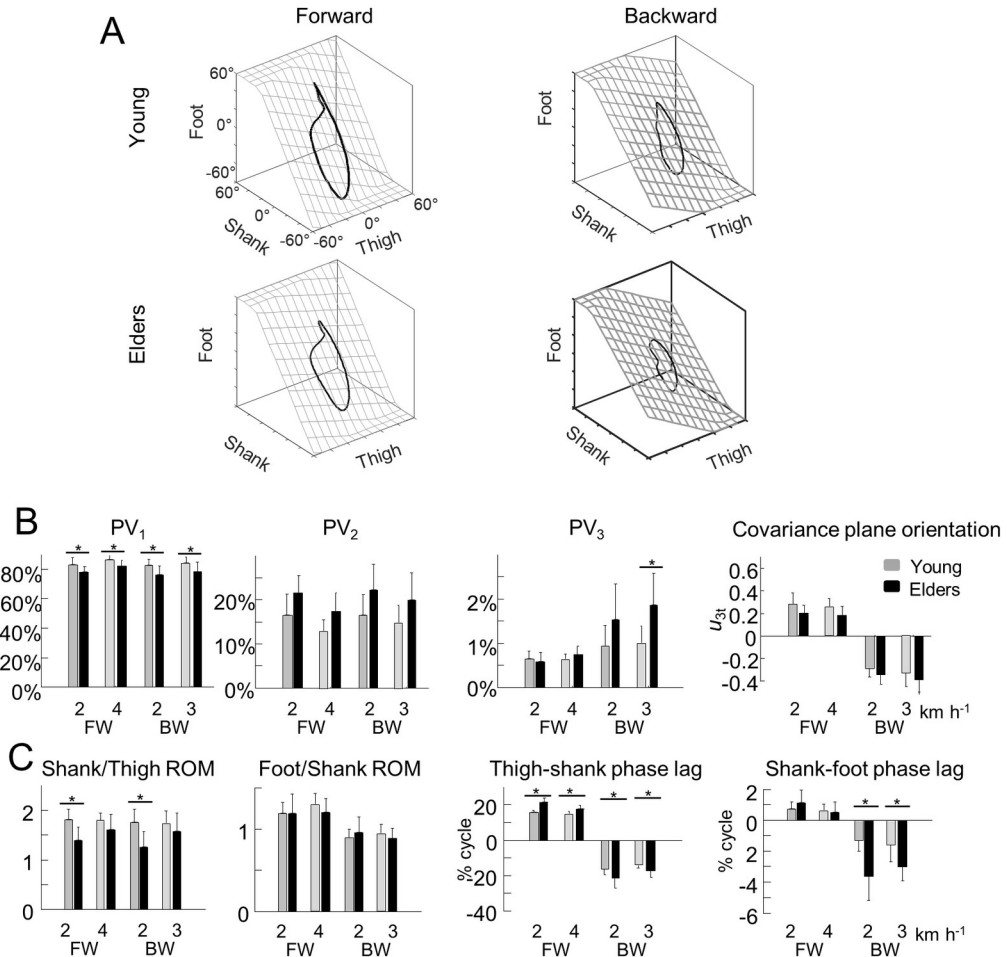

**Fig 2. Planar covariation of elevation angles.** *A*–Covariation of the ensemble-average limb-segment elevation angles during forward (right—1.11 m s⁻¹) and backward (left– 3 km h⁻¹) walking in young (up) and older adults (bottom). Note that when the elevation angles of thigh, shank and foot are plotted one versus the other in a x-y-z space, they co-vary along a loop constrained on a plane (x–y). Grids show the best-fitting plane. *B*–Average percentage of variance accounted for by the first (left—PV₁), second (middle—PV₂) and third (right–PV₃) eigenvector of the principal component analysis and the direction cosines of the normal to the covariation plane with the positive semi-axis of the thigh angular coordinates (u3t). *C*–Average amplitude ratios between the range of motion of adjacent segments and phase lags between time curves of elevation angles of adjacent segments. In panel A and B, grey bars correspond to young adults, whereas black bars correspond to older adults. The * indicates a significant effect of age.

20.9; $p < 0.001$; $\eta_p^2 = 0.24$). Furthermore, PV₁ slightly but significantly increased (and PV₂ decreased) with speed (PV₁: $F_{2,67} = 4.7$; $p = 0.012$; PV2: $F_{2,67} = 4.9$; $p = 0.010$), but was not significantly affected by the walking direction (PV₁: $F_{1,67} = 0.4$; $p = 0.534$; PV2: $F_{1,67} = 0.6$; $p = 0.802$). The percentage of variance accounted for by the third PC (PV₃), which represents the deviation from planarity, did not change significantly with speed ($F_{2,67} = 1.4$; $p = 0.259$), but increased significantly during backward walking ($F_{1,67} = 13.2$; $p = 0.002$) and in older adults ($F_{1,67} = 13.2$; $p = 0.001$; $\eta_p^2 = 0.16$). In addition, the effect of walking direction was significantly greater in older adults (interaction: $F_{1,67} = 4.4$; $p = 0.039$).

Obviously, during the stance phase of backward walking the foot relative to the hip moves from back to front, whereas in forward walking the foot moves from front to back. Accordingly, the orientation of the loop formed by the thigh, shank and foot elevation angles is reversed during backward walking as compared to forward walking [36], resulting in an

opposite sign of the direction cosine $u_{3t}$ (Fig 2B; $F_{1,67} = 397.3$; $p < 0.001$). In both conditions, $u_{3t}$ was significantly smaller in older than in young adults ($F_{1,67} = 8.2$; $p = 0.006$; $\eta_p^2 = 0.12$) but was not significantly affected by the speed of progression ($F_{2,67} = 1.2$; $p = 0.295$).

Both the shape of the loop and the orientation of the plane depend on the amplitude ratio and the time relationship characteristics of adjacent elevation angles (Fig 2C). The amplitude ratio between thigh and shank segments was significantly smaller in older adults ($F_{1,67} = 21.1$; $p < 0.001$; $\eta_p^2 = 0.25$), but was not significantly affected by speed ($F_{2,67} = 2.1$; $p = 0.126$) or walking direction ($F_{1,67} = 1.0$; $p = 0.316$). The amplitude ratio between shank and foot segments was significantly smaller in backward walking ($F_{1,67} = 17.7$; $p < 0.001$), but was not significantly affected by speed ($F_{2,67} = 0.7$; $p = 0.507$) or age groups ($F_{1,67} = 0.9$; $p = 0.330$).

At each speed, the phase lags between adjacent segments were greatly affected by walking direction (thigh-shank: $F_{1,67} = 914.8$; $p < 0.001$; shank-foot: $F_{1,67} = 101.6$; $p < 0.001$): in forward walking, the phase lags were positive, showing that the oscillation of the proximal segment lead the distal ones, whereas in backward walking the phases lags were negative. In addition, the phase lags were significantly greater in older adults (thigh-shank: $F_{1,67} = 33.7$; $p < 0.001$; $\eta_p^2 = 0.31$; shank-foot: $F_{1,67} = 22.7$; $p < 0.001$; $\eta_p^2 = 0.24$), and the effect of age was significantly greater during backward walking (thigh-shank: $F_{1,67} = 19.5$; $p < 0.001$; shank-foot: $F_{1,67} = 16.8$; $p < 0.001$).

## EMG activities and spinal motor output

Fig 3A illustrates the ensemble averages of rectified EMG envelopes at all walking conditions in young and older adults. EMGs for forward walking were qualitatively consistent with those reported in the literature [28,68]. The activity patterns of some muscles for backward walking were strikingly different from those of forward walking. For example, *BF* and *ST* were mostly active during early stance and the end of swing in forward walk, whereas they were mostly active during early stance in backward walk. The ankle extensors *GM*, *GL*, *SOL* and *PERL* were mostly active during mid-stance in forward walking, whereas they were active during late swing and early stance in backward walking.

In older adults, the EMG data remained roughly similar to young adults. However, some muscles were characterized by a different duration of activation, which we estimated as the FWHM (Fig 3B). In particular, the trunk extensor muscles (*ES*: $F_{1,59} = 12.4$; $p = 0.001$; $\eta_p^2 = 0.17$), the hamstrings (*BF*: $F_{1,61} = 10.0$; $p = 0.002$; $\eta_p^2 = 0.14$; *ST*: $F_{1,57} = 10.8$; $p = 0.002$; $\eta_p^2 = 0.16$), and the ankle extensors (*GM*: $F_{1,66} = 7.0$; $p = 0.010$; $\eta_p^2 = 0.09$; *GL*: $F_{1,59} = 9.4$; $p = 0.003$; $\eta_p^2 = 0.14$; *SOL*: $F_{1,61} = 39.9$; $p < 0.001$; $\eta_p^2 = 0.39$; *PERL*: $F_{1,56} = 9.7$; $p = 0.003$; $\eta_p^2 = 0.15$) presented significantly longer burst durations in older than in young adults.

Fig 4A presents the EMG of Fig 3A normalized to unit variance across all trials [60], mapped onto the estimated rostro-caudal location of the MN pools in the spinal cord (see Methods). The lumbar segments showed one major spot of activity around touchdown, involving primarily hip and knee extensors, whereas the sacral segments showed one major spot of activity around lift-off, mainly corresponding to the ankle extension at the end of stance [65,69,70]. The burst timing and duration of the spinal segments differed with age and direction of progression. As compared to young adults, the FWHM of the sacral MN activation was significantly greater in older adults ($F_{1,65} = 19.9$; $p < 0.001$; $\eta_p^2 = 0.23$), whereas FWHM of the lumbar MN activation was not significantly different ($F_{1,65} = 2.9$; $p = 0.091$; Fig 4). In addition, the occurrence of the maximal activation of the lumbar segment occurred significantly earlier in older than in young adults ($F_{1,65} = 11.1$; $p = 0.001$; $\eta_p^2 = 0.14$). Instead, no significant difference with age was observed at the sacral level ($F_{1,65} = 1.1$; $p = 0.307$), except for an earlier activation in backward than forward walking ($F_{1,65} = 40.3$; $p < 0.001$). In addition, the effect of walking direction was significantly greater in older adults (interaction: $F_{1,65} = 7.4$; $p = 0.008$).

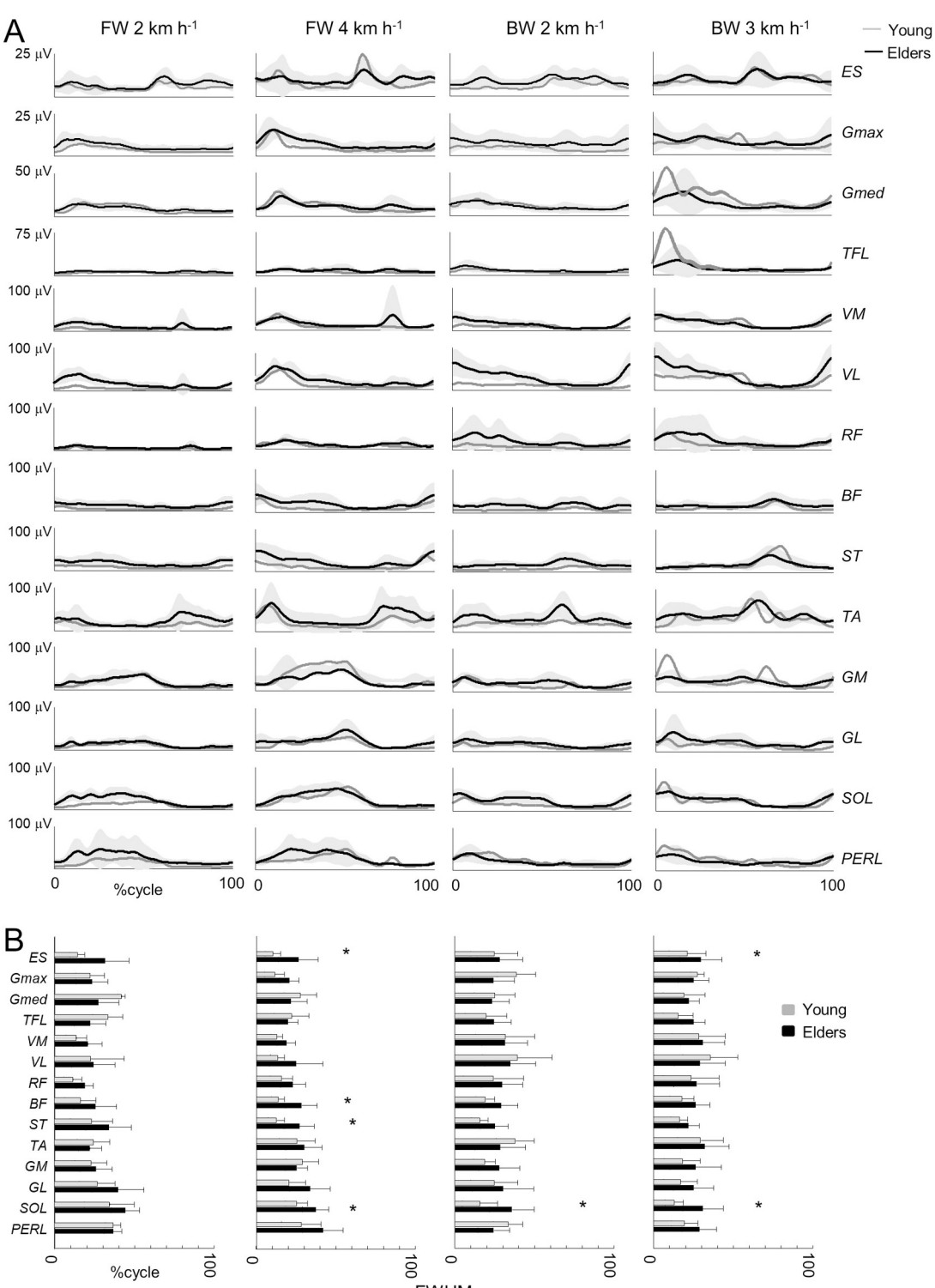

**Fig 3. Ensemble-averaged electromyogram (EMG) patterns during forward and backward walking in young and older adults.** *A*–ensemble-averaged EMG patterns over one stride. *ES*, erector spinae; *GM*, gluteus maximus; *Gmed*, gluteus medius; *SART*, sartorius; *TFL*, tensor fascia latae; *ADD*, adductor longus; *VM*, vastus medialis; *VL*, vastus lateralis; *RF*, rectus femoris; *BF*, biceps femoris; *ST*, semitendinous; *TA*, tibialis anterior; *MG*, gastrocnemius medialis; *LG*, lateral gastrocnemius; *SOL*, soleus; *PERL*, peroneus longus. The curves presented here are the average of the mean-curves of all the young (grey lines) and older (black lines) adults. The grey zone represents±1 SD for the older adults. *B*–Full Width Half Maximum (FWHM) of the 14 lower-limb muscles at each walking condition. The bars represent the grand mean of all the young (grey) and the older (black) adults. Thin lines represent one standard deviation. The * indicates a significant effect of age.

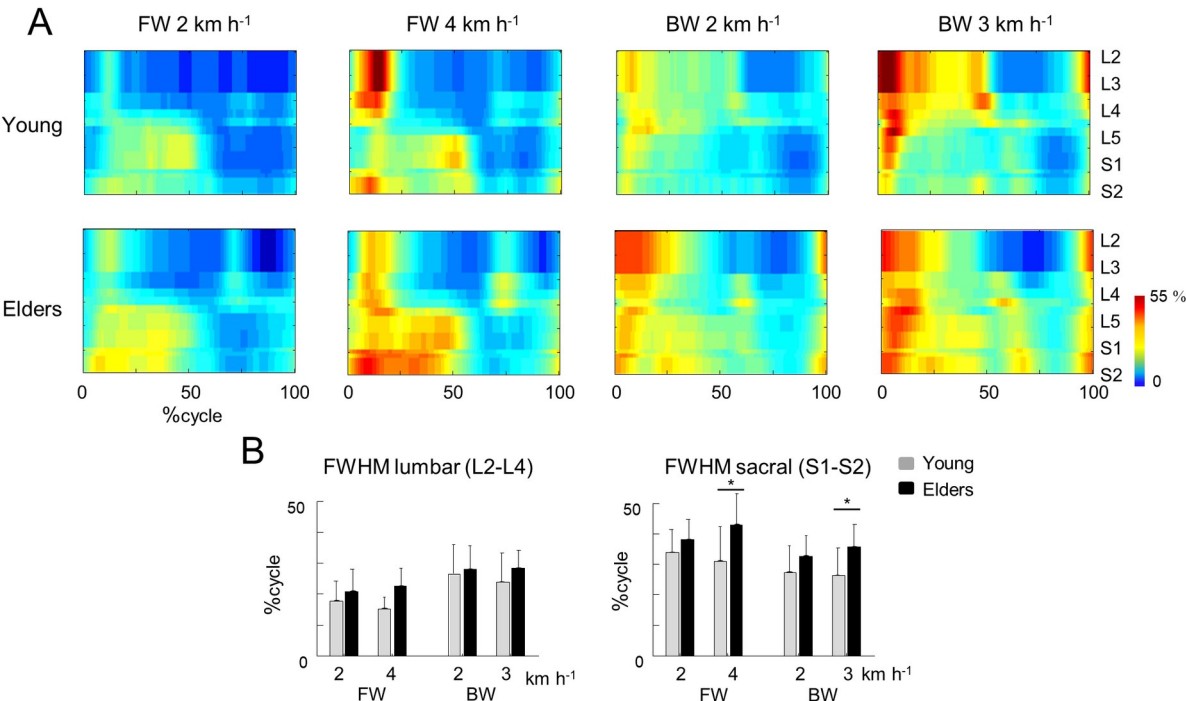

**Fig 4.** Spatiotemporal spinal motor outputs computed from normalized EMGs (A) and average full width half maximum and mean activation of the lumbar (top) and sacral (bottom) segments (B) during forward and backward walking. For each individual, EMG signals from each muscle were normalized to unit variance across all trials [60]. The bars represent the grand mean of all the young (grey) and the older (black) adults. Thin lines represent one standard deviation. The * indicates a significant effect of age.

Another age-related difference was represented by the intensities of the sacral and lumbar segments, when evaluated from non-normalized EMGs (Fig 3A and S1 Fig). The sacral mean intensity increased significantly with speed ($F_{2,67}$ = 5.1; $p$ = 0.009), whereas the lumbar mean intensity did not change significantly with speed ($F_{2,67}$ = 2.15; $p$ = 0.124). Reversing the walking direction augmented the engagement of lumbar segments ($F_{1,65}$ = 5.3; $p$ = 0.025), without affecting the sacral ones ($F_{1,65}$ = 0.5; $p$ = 0.470). Older adults presented significantly greater intensities at both lumbar and sacral levels (lumbar: $F_{1,65}$ = 18.3; $p$< 0.001; $\eta_p^2$ = 0.21; sacral: $F_{1,65}$ = 8.3; $p$ = 0.005; $\eta_p^2$ = 0.11). Notice that the effect of age on the burst duration did not depend on normalization (S1 Fig): the FWHM of the sacral MN activation was greater in older adults ($F_{1,65}$ = 19.4; $p$< 0.001; $\eta_p^2$ = 0.23), whereas FWHM of the lumbar MN activation was not different ($F_{1,65}$ = 1.8; $p$ = 0.180).

## Discussion

In this study, we investigated the effect of aging on the neuromuscular control of both forward and backward walking. Changing direction of locomotion is performed rather readily by both young and older adults. The effect of age on forward gait pattern has been often associated with a reduction of ankle propulsion force, for instance [19]. Despite the inverted plantigrade–digitigrade sequence during backward walking, we found similar age-related modifications of kinematic coordination and muscle activities in both backward and forward walking, suggesting specific adjustments of the motor control. In addition, we found that the age-related modifications on the intersegmental coordination were greater during backward walking.

With aging, motor weakness is due in part to neuromuscular degeneration, but also to degenerative changes in the central nervous system. Thus, reduction in grey matter volume

[71], number of motor cortical [72] and spinal motor neurons [73], synaptic density [74], white matter integrity [75], and descending commands for motor activation [76] are some of the factors that may contribute to age-related motor impairment.

To date, several studies have dealt with the effects of aging on forward walking. Our results are aligned with prior results showing that older subjects take shorter steps [77] (Fig 1) and adapt their intersegmental coordination mainly by changing the amplitude and phase of shank and foot motion [1,13,21,22] (Fig 2A). Most authors of previous work discuss a reduction in mechanical power generated by the plantarflexor muscles as the hallmark biomechanical features of older gait. More recently, it has also been shown that older adults display longer bursts of muscle activation [29] (Fig 3) that could be related to a more robust neuromuscular control (i.e. more able to cope with errors) to deal with poorer balance control [78,79]. Again, the reduced dynamic stability in older adults has been associated with a diminished ankle push-off [18].

Here, we found that the modifications of the intersegmental coordination during backward walking are similar to those during forward walking. In particular, the changes in the orientation of the covariation plane with age (Fig 2B) are mainly related to a change of the phase shift between shank and foot elevation angles (Fig 2C; [1]). The more in-phase oscillation of the shank and the foot in older adults may explain the reduction of ankle ROM with aging (Fig 1B), which is not only due to shorter steps (Fig 1C) since the ratio between proximal and distal segments also changed (Fig 2C). This reduced angular excursion at the ankle in older adults has already been ascribed to co-contractions of distal antagonist muscles, in part due to EMG widening [58]. Accordingly, the activity profiles of the muscles innervated by the sacral segments were significantly wider in older adults (Fig 4A). The reconstructed spinal maps of MN activity further illustrate this finding. Similar results during forward walking at matched cadence were previously documented by Monaco et al. [28], suggesting that the widening of EMG is not dependent on spatiotemporal gait parameters. In addition, this result does not simply reflect the documented distal-to-proximal modification of kinematics or kinetics, since the human spinal topography does not reflect the muscle topography on the lower limbs. Indeed, both distal (*GM*, *GM*, *SOL*, *PERL*) and proximal (*BF*, *ST*) muscles mainly innervated by distal segments of the spinal cord [55,80] displayed wider activations (Fig 3B).

The present results of a caudal-cranial gradient of involvement of the spinal locomotor segments in older adults remain to be explained. It is well established that there exists a cranio-caudal gradient of corticospinal development in infancy [81], but less is known about differential degeneration of different portions of the corticospinal tract with aging. In general, it appears that projection tracts, such as the corticospinal tract, which develop earlier than association tracts in infancy, degenerate later than association tracts in older subjects [82].

Normal aging and the development of neurodegeneration are two processes that are closely linked [83,84]. Indeed with aging, neurodegeneration might occur when cells fail to adapt to the increases in oxidative, metabolic and ionic stress [85]. In addition, the disks between the vertebrae become hard and brittle when aging. As a result, more pressure is put on the spinal cord and on the spinal nerve roots, especially in distal segments. These observations may partly underlie the distal-to-proximal age-related changes in the neuromuscular system with aging. Accordingly, when the spinal excitability is estimated using the Hoffmann reflex technique, no difference is found between young and older adults on *VM* muscle [86], whereas age-related modulations of the reflex response have been reported in *SOL* muscle [87]. Further, during standing, when a perturbation is delivered, older adults exhibit intermittent reversals of the classical distal-proximal postural synergy used in young adults [88]. Similarly, proximal muscles tend to be activated first in the paretic limb of hemiplegic subject [89]. Interestingly, Martino et al. [90] found that the spinal maps of patients affected by hereditary spastic paraplegia were characterized by a spread of the loci of activation at the sacral segments and, at more

severe stages, the lumbar segments, somewhat reminiscent of what happens in older adults. It is theoretically possible that the age-related changes in the neuromuscular control of gait are, at least in part, related to the progression of the aging degenerative process within the corticospinal tract, involving initially the sacral segments and later the lumbar segments. However, this possibility must be corroborated by studies specifically investigating changes in corticospinal innervation of different spinal segments.

The fact that the age-related modifications of neuromuscular control of gait observed during forward walking are also observed during backward walking indirectly supports the idea that walking impairment is not solely dependent on the reduction of force generated by the plantar-flexor muscles. Indeed, during backward walking, plantar flexion plays only a small role in propulsion [91]. The similarity of age-related modifications between the two walking directions indirectly supports the idea that somewhat similar spinal automatisms are used for forward and backward walking, as proposed by Grasso et al [36], Earhart et al. [92] and Ivanenko et al. [38], with a partial reconfiguration of lower-level networks [39] plus the probable intervention of supraspinal elements specifically for backward walking [40,41].

On the other hand, it has been shown that the plasticity associated with locomotor adaptation in human is direction specific, suggesting separate functional networks controlling forward and backward walking [54]. Moreover, backward walking is more challenging for the nervous system [40,41], and this style of walking is much less practiced than forward walking. Accordingly, we expected that the age-related differences of the neuromuscular control of gait would be greatly evidenced during backward walking. Indeed, several findings support the idea that backward walking may unmask mobility impairments in adult stroke patients [43], Parkinson disease [93], and in children with cerebral palsy [42]. By comparing older to young adults, greater adjustments of spatiotemporal gait parameters have been observed during backward than forward walking [2,31,52]. However, in these studies, subjects walked at self-selected speed and their velocity was significantly lower for older adults. In addition, the reduction of velocity was greater in backward than forward walking, and it was therefore difficult to differentiate the effect of age from the effect of speed.

By comparing young and older adults at matched speeds, we showed an interaction between age group and the direction of progression on the relative duration of the stance phase (Fig 1C), on $PV_3$ (Fig 2B), on the phase lags between adjacent segment (Fig 2C), and on the timing of maximal sacral MN activity. By separating the effects of concomitant issues, such as age and speed, these changes of gait parameters, kinematics and muscle activity suggest that older persons have greater deficits in backward performance than during forward walking.

Our sample size was relatively small and the older subjects tended to be active and had no recent history of falls. Future studies involving a more heterogeneous population of individuals should be designed to focus on specific gait abnormalities in challenging conditions as a function of physical functioning. It is also important to note that the familiarization time for the backward walking task was limited. Whether similar differences between age groups would be still apparent after longer familiarization to backward walking is also an open question. Nevertheless, the findings of this study extend the available information on age-related differences in the neuromuscular control of gait occurring both during backward and forward walking. In addition, the results suggest that assessing backward walking in clinical practice may shed light on or even unmask neuromuscular adjustments of gait in older adults.

## Supporting information

**S1 Fig. Unilateral spatiotemporal spinal motor outputs computed from ensemble-averaged electromyograms (EMGs) during forward and backward walking.** *A*—Ensemble-

averaged normalized electromyogram (EMG) patterns. For each individual, EMG signals from each muscle were normalized to unit variance across all trials. *B*–Motor output (reported in μV) is plotted as a function of gait cycle in young (top) and older (bottom) adults. *C*–Average full width half maximum and mean activation of the lumbar (top) and sacral (bottom) segments. The bars represent the grand mean of all the young (grey) and the older (black) adults. Thin lines represent one standard deviation. The * indicates a significant effect of age. (TIF)

**S1 Table.** List of muscles analysed (1) or removed (0) for each condition (from left to right: Forward 2 km h⁻¹; Forward 4 km h⁻¹; Backward 2 km h⁻¹; Backward 3 km h⁻¹) in young (Y) and older (O) adults.
(DOCX)

**S2 Table. Average (mean± SD) coefficient of correlation *r* between the activation of each individual spinal segment (from L2 to S2) reconstructed from a subset of seven muscles (minimum number of muscle recorded) and from the full set of muscles in each walking condition.**
(DOCX)

**S1 Data.**
(XLS)

## Author Contributions

**Conceptualization:** Arthur H. Dewolf, Yury Ivanenko, Francesco Lacquaniti.

**Data curation:** Arthur H. Dewolf, Francesca Sylos-Labini, Germana Cappellini, Yury Ivanenko.

**Formal analysis:** Arthur H. Dewolf, Yury Ivanenko, Francesco Lacquaniti.

**Funding acquisition:** Yury Ivanenko, Francesco Lacquaniti.

**Investigation:** Arthur H. Dewolf, Yury Ivanenko, Francesco Lacquaniti.

**Methodology:** Arthur H. Dewolf, Yury Ivanenko, Francesco Lacquaniti.

**Project administration:** Yury Ivanenko, Francesco Lacquaniti.

**Resources:** Francesco Lacquaniti.

**Software:** Arthur H. Dewolf.

**Supervision:** Yury Ivanenko, Francesco Lacquaniti.

**Validation:** Arthur H. Dewolf, Yury Ivanenko, Francesco Lacquaniti.

**Visualization:** Arthur H. Dewolf, Francesco Lacquaniti.

**Writing – original draft:** Arthur H. Dewolf.

**Writing – review & editing:** Arthur H. Dewolf, Francesca Sylos-Labini, Germana Cappellini, Yury Ivanenko, Francesco Lacquaniti.

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
