## [Decision Letter · Decision Letter 0]

9 Dec 2020

PONE-D-20-34850

Age-related changes in the neuromuscular control of forward and backward locomotion

PLOS ONE

Dear Dr. Dewolf,

Thank you for submitting your manuscript to PLOS ONE. After careful consideration, we feel that it has merit but does not fully meet PLOS ONE’s publication criteria as it currently stands. Therefore, we invite you to submit a revised version of the manuscript that addresses the points raised during the review process.

While both reviewers are positive, both suggested to improve the clarity, mainly in discussion.  Please follow each's suggestions.

We look forward to receiving your revised manuscript.

Kind regards,

Kei Masani

Academic Editor

PLOS ONE

Journal Requirements:

"All subjects gave their informed consent. Experiments were performed according to the Declaration of Helsinki and were approved by the local ethics committee CE/PROG749."

a.) Please amend your current ethics statement to include the full name of the ethics committee/institutional review board(s) that approved your specific study.

b.) Please provide additional details regarding participant consent.  In the ethics statement in the Methods and online submission information, please ensure that you have specified what type you obtained (for instance, written or verbal, and if verbal, how it was documented and witnessed). If your study included minors, state whether you obtained consent from parents or guardians. If the need for consent was waived by the ethics committee, please include this information.

c.) Once you have amended this/these statement(s) in the Methods section of the manuscript, please add the same text to the “Ethics Statement” field of the submission form (via “Edit Submission”).

Reviewers' comments:

Reviewer's Responses to Questions

**Comments to the Author**

1. Is the manuscript technically sound, and do the data support the conclusions?

Reviewer #1: Yes

Reviewer #2: Partly

2. Has the statistical analysis been performed appropriately and rigorously? 

Reviewer #1: Yes

Reviewer #2: Yes

3. Have the authors made all data underlying the findings in their manuscript fully available?

Reviewer #1: No

Reviewer #2: No

4. Is the manuscript presented in an intelligible fashion and written in standard English?

Reviewer #1: Yes

Reviewer #2: Yes

5. Review Comments to the Author

Reviewer #1: In this simply designed, yet informative study, the authors investigated kinematic and electromyographic modifications to the gait pattern imputable to aging and walking direction and speed. By using well-established techniques, some of which they came up with in the first place, they nicely presented the outcomes and discussed them leveraging on a good mixture of classical and recent neuroscience/biomechanics literature. The results are certainly convincing, but the discussion could be a little stronger, especially in the paragraph discussing the widening of EMG signals and their spinal motoneuron mapping.

MINOR COMMENTS

1. I am not a native English speaker. However, in science and communication at large it is nowadays fairly common to avoid the use of the term “elderly” and replace it with “older adults”, “older persons”, “aging adult”, or similar.

2. A few typos are sparsely present in the manuscript and it would be nice if the authors could go once more through the text to solve this minor but unaesthetic issue.

3. Lines 25-27: this opening sentence in the abstract sounds a bit off. Maybe the authors could consider rephrasing?

4. Line 58: “muscular skeletal” or perhaps “musculoskeletal”?

5. Line 123: I believe that the name of the institution responsible for the organisation of the Ethics Committee should be reported here.

6. Line 125: with “selected speed” do the authors mean “fixed speed” or similar? This made me stop for a second.

7. Lines 144-145: here the authors state that some electrodes were removed from the analysis. They possibly meant that the data series produced by those channels were excluded, but there is no mention of how they dealt with missing data. Were those time series simply fed into the statistics as vectors of NAs? It would be nice to read a brief statement on this statistical matter.

8. Line 198: checking for the normality of residuals is certainly one of those debated things in statistics. However, there exist some tests (e.g. the Shapiro-Wilk test comes immediately to mind) that might give a more objective outcome than the “eyeballing” reported by the authors. Here I would recommend, if feasible, to refrain from visual check of normality.

9. Line 243: I did not find any Figure 2D.

10. Fig. 3A: to what are those average EMG normalised? It is a shame that may proximal muscles have such low and unreadable signals (while it looks like there indeed was some activity that is simply not to be clearly seen). Would it be possible for the authors to improve readability? An idea could be to normalise each muscle to the maximum of each condition or to just rescale the ordinates. Just a suggestion.

11. Lines 317-344: I felt that a strong, final message is missing here. The authors argue in favour of the well-known distal/proximal plasticity distribution, correctly admitting that their results showed increased FWHM also in muscles that are not innervated by sacral segments (ES being mostly innervated by thoracic segments and the hamstring having a non-negligible source in the lower lumbar segments, at least ST). Yet, I missed the conclusion. This does not necessarily mean that the authors did not write it, but I just could not see it. Here I would warmly suggest some reorganisation.

Reviewer #2: This study addressed to investigate age-related changes in gait kinematics and multiple muscle activities by comparing forward and backward treadmill walking between young and elderly adults. Principal component analysis applying to lower limb joint angles and spatiotemporal mapping of multiple lower limb muscle activities onto spinal location revealed that the joint coordination and mapping patterns were changed in elderly adults particularly during backward walking.

Richness of results about various gait-related parameters provides diverse understanding of age-related changes in locomotor control. In addition, the methodology to assess the gait kinematics and muscle activities is sophisticated, and the results derived from the approach are sound. However, it wasn’t clear to me what are the main findings of this study, and therefore, which are the novel findings of this study. There has been a lot of related works; and indeed, the several present results were aligned with prior results especially in a forward gait. I think the authors need to clearly articulate the specific gait parameters newly focused in the present study.

Furthermore, I and perhaps a lot of readers have a strong interest in the relationship among the age-related changes in gait kinematics, muscle activities and their mapping onto spinal location. However, the relationship is not clarified, which might also create ambiguity of the main results. I strongly recommend the authors at least discuss the involvement of the age-related gait parameters clarified in this study to promote the comprehensive understanding of age-related changes in locomotor control.

Specific comments are described below.

Line 66-67: I’m confused with “the underuse of ankle muscles”. Does it denote the reduced effort of an ankle joint?

Line 73-75: It is hard to understand novelty of the problem that the authors point out as compared with the previous study (Monaco et al., 2010, J Neurophysiol.).

Line 102-105: I think the authors need to clarify the gait parameters described here.

Line 108-110: It wasn’t clear how the age-related differences of neuromuscular control is important during backward walking.

Line 116-120: While the significant gait parameters are joint coordination and spatiotemporal muscle activities, why was the total sample size determined based on the age-related difference on stride length during forward walking? The authors need to describe the reason why the sample size was determined to be sufficient.

Line 144-145 and Table 1: My concern is substantial removal of EMG electrodes from the analysis. The authors need to verify the removal of some electrodes doesn’t affect the results, especially in a spinal map. Furthermore, a table that shows what electrodes (muscles) were removed should be added.

Line 165-166: I think joint angle data should be normalized to have zero mean and unit variance before applying a principal component analysis to avoid estimates of the joint coordination biased to a particular joint.

Line 243-246: A brief interpretation about the change in the direction cosine helps understanding of the result of the joint coordination.

Line 277-288: As well as the results of joint coordination, it is hard to interpret what the difference in mapping patterns between young and elderly adults means. A brief description at least about muscles innervated from a significant spinal location will promote understanding of the results and significance of the spinal mapping.

Line 307: “ageing”  “aging”

Line 366: “the effect age” probably should be modified to “the effect of age”.

I think the aspect of different functional network controlling forward and backward walking (Choi and Bastial, 2007, Nat Neurosci.) will develop the discussion about the age-related modification specific to the walking direction.

I hope these comments will be helpful.

6. PLOS authors have the option to publish the peer review history of their article (what does this mean?). If published, this will include your full peer review and any attached files.

Reviewer #1: **Yes: **Alessandro Santuz

Reviewer #2: No

---

## [Author Response · Author response to Decision Letter 0]

28 Dec 2020

Review Comments to the Author

Reviewer #1: 

In this simply designed, yet informative study, the authors investigated kinematic and electromyographic modifications to the gait pattern imputable to aging and walking direction and speed. By using well-established techniques, some of which they came up with in the first place, they nicely presented the outcomes and discussed them leveraging on a good mixture of classical and recent neuroscience/biomechanics literature. The results are certainly convincing, but the discussion could be a little stronger, especially in the paragraph discussing the widening of EMG signals and their spinal motoneuron mapping.

We would like to thank the reviewer for providing valuable and constructive comments and suggestions that have helped us to improve the manuscript. We have responded to all of the points raised and revised the manuscript accordingly.

MINOR COMMENTS

1. I am not a native English speaker. However, in science and communication at large it is nowadays fairly common to avoid the use of the term “elderly” and replace it with “older adults”, “older persons”, “aging adult”, or similar.

After a quick check, we discovered that the Journal of Geriatric Physical Therapy suggested that the term older adult or older person is respectful and should be the standard term (Dale et al., 2011 – Use of the Term “Elderly”). Therefore, we now use ‘older adults’ throughout the manuscript. 

2. A few typos are sparsely present in the manuscript and it would be nice if the authors could go once more through the text to solve this minor but unaesthetic issue.

Sorry for that. We have fixed that.

3. Lines 25-27: this opening sentence in the abstract sounds a bit off. Maybe the authors could consider rephrasing?

Accordingly, we have changed the first sentence of the abstract. Now, it reads: 

‘Previous studies found significant modification in spatiotemporal parameters of backward walking in healthy older adults, but the age-related changes in the neuromuscular control have been considered to a lesser extent.’

4. Line 58: “muscular skeletal” or perhaps “musculoskeletal”?

Correct. Thanks.

5. Line 123: I believe that the name of the institution responsible for the organisation of the Ethics Committee should be reported here.

Done.

6. Line 125: with “selected speed” do the authors mean “fixed speed” or similar? This made me stop for a second.

Indeed, we’ve changed it into ‘fixed speeds’.

7. Lines 144-145: here the authors state that some electrodes were removed from the analysis. They possibly meant that the data series produced by those channels were excluded, but there is no mention of how they dealt with missing data. Were those time series simply fed into the statistics as vectors of NAs? It would be nice to read a brief statement on this statistical matter.

Indeed, the removed muscles were replaced by a Not-a-number vector. We have added this information in the Methods section. We have also added a Table (S1), showing what electrodes (muscles) were removed.

 ‘In certain conditions, some electrodes became partially detached and the data series produced by these electrodes were removed from the analysis (replaced by a not-a-number vector) on a subject-specific basis (Table S1).’

‘Note that, consistent with previous work [64], the spinal maps were relatively insensitive to the subset of muscles analysed (Table 1). Indeed, spinal maps reconstructed from a subset of seven muscles (minimum number of muscles recorded) were strongly correlated with the maps computed from the full set of muscles, with average correlation coefficients between 0.9–0.99 for each task and at each individual spinal segment (Table S2).’

8. Line 198: checking for the normality of residuals is certainly one of those debated things in statistics. However, there exist some tests (e.g. the Shapiro-Wilk test comes immediately to mind) that might give a more objective outcome than the “eyeballing” reported by the authors. Here I would recommend, if feasible, to refrain from visual check of normality.

We agree with the reviewer. In the revised version, the normality of the residuals was checked by the Kolmogorov-Smirnov test. Normality were not assumed for three additional variables. We have updated the results, but fortunately, significant results remain significant and vice versa.

9. Line 243: I did not find any Figure 2D.

Thanks, we referred to Fig. 2B.

10. Fig. 3A: to what are those average EMG normalised? It is a shame that may proximal muscles have such low and unreadable signals (while it looks like there indeed was some activity that is simply not to be clearly seen). Would it be possible for the authors to improve readability? An idea could be to normalise each muscle to the maximum of each condition or to just rescale the ordinates. Just a suggestion.

In Fig 3A, the EMG presented were not normalized (in �V). As suggested, we have changed the scale of the proximal muscles. It already improves the readability. In addition, we have also added normalized EMG in the Fig. S1.

11. Lines 317-344: I felt that a strong, final message is missing here. The authors argue in favour of the well-known distal/proximal plasticity distribution, correctly admitting that their results showed increased FWHM also in muscles that are not innervated by sacral segments (ES being mostly innervated by thoracic segments and the hamstring having a non-negligible source in the lower lumbar segments, at least ST). Yet, I missed the conclusion. This does not necessarily mean that the authors did not write it, but I just could not see it. Here I would warmly suggest some reorganisation.

Thanks. We have reorganized the paragraph in order to clarify its message. Now, it reads: ‘[…] Interestingly, Martino et al. [90] found that the spinal maps of patients affected by hereditary spastic paraplegia were characterized by a spread of the loci of activation at the sacral segments and, at more severe stages, the lumbar segments, somewhat reminiscent of what happens in older adults. It is theoretically possible that the age-related changes in the neuromuscular control of gait are, at least in part, related to the progression of the aging degenerative process within the corticospinal tract, involving initially the sacral segments and later the lumbar segments. However, this possibility must be corroborated by studies specifically investigating changes in corticospinal innervation of different spinal segments.

’

Reviewer #2:

This study addressed to investigate age-related changes in gait kinematics and multiple muscle activities by comparing forward and backward treadmill walking between young and elderly adults. Principal component analysis applying to lower limb joint angles and spatiotemporal mapping of multiple lower limb muscle activities onto spinal location revealed that the joint coordination and mapping patterns were changed in elderly adults particularly during backward walking.

Richness of results about various gait-related parameters provides diverse understanding of age-related changes in locomotor control. In addition, the methodology to assess the gait kinematics and muscle activities is sophisticated, and the results derived from the approach are sound. However, it wasn’t clear to me what are the main findings of this study, and therefore, which are the novel findings of this study. There has been a lot of related works; and indeed, the several present results were aligned with prior results especially in a forward gait. I think the authors need to clearly articulate the specific gait parameters newly focused in the present study.

Furthermore, I and perhaps a lot of readers have a strong interest in the relationship among the age-related changes in gait kinematics, muscle activities and their mapping onto spinal location. However, the relationship is not clarified, which might also create ambiguity of the main results. I strongly recommend the authors at least discuss the involvement of the age-related gait parameters clarified in this study to promote the comprehensive understanding of age-related changes in locomotor control.

We would like to thank the reviewer for the interesting comments that were helpful. We have responded to all of the points raised and revised the manuscript accordingly. 

Among others, we have reorganized the discussion, and added in the discussion section links between the age-related changes in gait kinematics, muscle activities. Please see

‘Here, we found that the modifications of the intersegmental coordination during backward walking are similar to those during forward walking. In particular, the changes in the orientation of the covariation plane with age (Fig. 2B) are mainly related to a change of the phase shift between shank and foot elevation angles (Fig. 2C; [1]). The more in-phase oscillation of the shank and the foot in older adults may explain the reduction of ankle ROM with aging (Fig. 1B), which is not only due to shorter steps (Fig. 1C) since the ratio between proximal and distal segments also changed (Fig. 2C). This reduced angular excursion at the ankle in older adults has already been ascribed to co-contractions of distal antagonist muscles, in part due to EMG widening [58]. Accordingly, the activity profiles of the muscles innervated by the sacral segments were significantly wider in older adults (Fig. 4A). The reconstructed spinal maps of MN activity further illustrate this finding (Fig. 5A). Similar results during forward walking at matched cadence were previously documented by Monaco et al. [28], suggesting that the widening of EMG is not dependent on spatiotemporal gait parameters.’

Specific comments are described below.

Line 66-67: I’m confused with “the underuse of ankle muscles”. Does it denote the reduced effort of an ankle joint?

Franz [19] showed that many old adults underutilize their available muscular capacity for generating propulsive power in walking, suggesting that factors other than muscle weakness contribute to the age-associated reduction in propulsive power generation, and in turn in the age-related modifications of gait. We have clarified this idea, and it reads now: ‘The decline of propulsive power generation during push-off is thus not only due to a reduced muscular capacity but might also emerge from a different neuromuscular control strategy [19]..’

Line 73-75: It is hard to understand novelty of the problem that the authors point out as compared with the previous study (Monaco et al., 2010, J Neurophysiol.).

In this paragraph, we detailed the published data on forward walking. We have decided to remove the second part of the sentence related to the difference between the present study and the published work of Monaco and collaborators (since the major novelty is backward walking condition).

Note that we previously referred to the fact that, even if Monaco et al. (2010) correctly visually identified that ‘bursts of MN activity occurred likely than in young even though they were characterized by higher amplitude, and […] they were more spread’, these authors did not quantified it. In addition, they also compared young and older adults at matched cadence whereas in the present study we matched the speeds. We have now included this information in the Discussion section. 

Please, see: ‘Similar results during forward walking at matched cadence were previously documented by Monaco et al. [28], suggesting that the widening of EMG is not dependent on spatiotemporal gait parameters.’

Line 102-105: I think the authors need to clarify the gait parameters described here. Line 108-110: It wasn’t clear how the age-related differences of neuromuscular control is important during backward walking.

Thanks. We have clarified our parameters at the end of the introduction. Please see Lines 100-110: ‘Altered spatiotemporal stride parameters [2,31], altered coordination patterns among the elevation angles of the lower limb segments [1,13], and wider bursts of muscle activity [28,29] have been previously documented for the forward locomotion of older adults. Here, we expected that some of these alterations might apply also to backward walking. In particular, we expected age-related adjustments of the intersegmental coordination, namely a more in-phase shank and foot motion, as well as a widening of muscle activities. Importantly, we also expected that some of these age-related modifications might be reflected in the pattern of rostrocaudal activation of the motoneuron pools. Finally, we hypothesized that these age-related differences of neuromuscular control would be more pronounced during backward walking compared with forward walking.’

Line 116-120: While the significant gait parameters are joint coordination and spatiotemporal muscle activities, why was the total sample size determined based on the age-related difference on stride length during forward walking? The authors need to describe the reason why the sample size was determined to be sufficient.

We computed the ‘expected effect’ from published findings from a similar study [1]. 

No data were available (mean + sd) concerning muscle widening and there is a documented interaction between the effect of speed and age on the intersegmental coordination of gait [1] (meaning that the ‘expected effect’ would considerably vary with walking speed). Therefore, we have decided to use general gait parameter to a priori determine our sample size. 

Line 144-145 and Table 1: My concern is substantial removal of EMG electrodes from the analysis. The authors need to verify the removal of some electrodes doesn’t affect the results, especially in a spinal map. Furthermore, a table that shows what electrodes (muscles) were removed should be added.

As suggested by the reviewer, we have also added a Table showing what electrodes (muscles) were removed. We have suggested publishing it as Supplementary materials (Table S1), because the table is quite long (4 conditions, 20 subjects, 14 muscles). Nevertheless, if the reviewer thinks that the table should be placed within the manuscript, it could be easily done.

In addition, we’ve also checked the sensitivity to the number of muscles analysed. To do that, we computed the spinal maps from all muscles we have (Table 1 & S1) and also from only seven muscles (minimum number of muscles recorded). We found that the spinal maps reconstructed from the reduced muscle subsets were strongly correlated with the maps computed from the full set of muscles, with average correlation coefficients between 0.9–0.99 for each task and at each individual spinal segment (Table S2). This information was added to the manuscript, and the Table added as Supplementary Materials.

Line 165-166: I think joint angle data should be normalized to have zero mean and unit variance before applying a principal component analysis to avoid estimates of the joint coordination biased to a particular joint.

In the present manuscript, a principal component analysis was applied to determine the covariance matrix of the segment elevation angles. As detailed in Ivanenko et al. (2008 – J Neurophy), the correlation matrix, i.e. the covariance matrix can be obtained after normalizing each segment to unit variance. While the eigenvalues of the covariance matrix do not depend on the plane orientation, pairwise correlations depend both on those eigenvalues and on the plane orientation. Thus, correlation is not an adequate measure of planarity in three dimensional spaces and, similarly, is not an adequate measure of subspace embedding in higher dimensional spaces.

We’ve now specified in the MS that we used non-normalized angles and have added the reference ([57]).

Line 243-246: A brief interpretation about the change in the direction cosine helps understanding of the result of the joint coordination.

We have clarified the sentence and add an interpretation in the Discussion section.

It reads now: ‘Obviously, during the stance phase of backward walking the foot relative to the hip moves from back to front, whereas in forward walking the foot moves from front to back. Accordingly, the orientation of the loop formed by the thigh, shank and foot elevation angles is reversed during backward walking as compared to forward walking [36], resulting in an opposite sign of the direction cosine u3t (Fig. 2B; F1,67= 397.3; p< 0.001).’

Line 277-288: As well as the results of joint coordination, it is hard to interpret what the difference in mapping patterns between young and elderly adults means. A brief description at least about muscles innervated from a significant spinal location will promote understanding of the results and significance of the spinal mapping.

Thanks. We have clarified it, as follow: ‘Figure 4A presents the EMG of Fig. 3A normalized to unit variance across all trials [60], mapped onto the estimated rostro-caudal location of the MN pools in the spinal cord (see Methods). The lumbar segments showed one major spot of activity around touchdown, involving primarily hip and knee extensors, whereas the sacral segments showed one major spot of activity around lift-off, mainly corresponding to the ankle extension at the end of stance [65,69,70].’

Line 307: “ageing”  “aging”

Changed. Thanks

Line 366: “the effect age” probably should be modified to “the effect of age”.

Done. Thanks

I think the aspect of different functional network controlling forward and backward walking (Choi and Bastial, 2007, Nat Neurosci.) will develop the discussion about the age-related modification specific to the walking direction.

Thank you for pointing this out. We agree that this paper should be discussed. 

First, we cited it in the Introduction section: 

‘In particular, because backward walking is more challenging than forward walking and because patterns of neuromuscular control are direction specific in humans [54], we wondered whether backward walking can reveal age-related modifications of gait that are not otherwise apparent during forward walking.’

In addition, we have added it in the Discussion section: 

‘On the other hand, it has been shown that the plasticity associated with locomotor adaptation in human is direction specific, suggesting separate functional networks controlling forward and backward walking [54]. Moreover, backward walking is more challenging for the nervous system [40,41], and this style of walking is much less practiced than forward walking.’

I hope these comments will be helpful.

---

## [Decision Letter · Decision Letter 1]

19 Jan 2021

Age-related changes in the neuromuscular control of forward and backward locomotion

PONE-D-20-34850R1

Dear Dr. Dewolf,

We’re pleased to inform you that your manuscript has been judged scientifically suitable for publication and will be formally accepted for publication once it meets all outstanding technical requirements.

Kind regards,

Kei Masani

Academic Editor

PLOS ONE

Additional Editor Comments (optional):

Reviewers' comments:

Reviewer's Responses to Questions

**Comments to the Author**

1. If the authors have adequately addressed your comments raised in a previous round of review and you feel that this manuscript is now acceptable for publication, you may indicate that here to bypass the “Comments to the Author” section, enter your conflict of interest statement in the “Confidential to Editor” section, and submit your "Accept" recommendation.

Reviewer #1: All comments have been addressed

Reviewer #2: All comments have been addressed

2. Is the manuscript technically sound, and do the data support the conclusions?

Reviewer #1: Yes

Reviewer #2: Yes

3. Has the statistical analysis been performed appropriately and rigorously? 

Reviewer #1: Yes

Reviewer #2: Yes

4. Have the authors made all data underlying the findings in their manuscript fully available?

Reviewer #1: No

Reviewer #2: Yes

5. Is the manuscript presented in an intelligible fashion and written in standard English?

Reviewer #1: Yes

Reviewer #2: Yes

6. Review Comments to the Author

Reviewer #1: I would like to thank the authors for their commitment to comply with all my requests. Congratulations on a very nice paper.

Reviewer #2: I thank the authors for responding to the reviewers' comments and then revising the manuscript. The manuscript has been much improved.

7. PLOS authors have the option to publish the peer review history of their article (what does this mean?). If published, this will include your full peer review and any attached files.

Reviewer #1: **Yes: **Alessandro Santuz

Reviewer #2: No

---

## [Editor Report · Acceptance letter]

21 Jan 2021

PONE-D-20-34850R1 

Age-related changes in the neuromuscular control of forward and backward locomotion 

Dear Dr. Dewolf:

I'm pleased to inform you that your manuscript has been deemed suitable for publication in PLOS ONE. Congratulations! Your manuscript is now with our production department. 

Kind regards, 

on behalf of

Dr. Kei Masani 

Academic Editor

PLOS ONE